# Electrophysiological and Behavioral Responses of *Holotrichia parallela* to Volatiles from Peanut

**DOI:** 10.3390/insects12020158

**Published:** 2021-02-13

**Authors:** Mengmeng Zhang, Zhihao Cui, Nuo Zhang, Guanglin Xie, Wenkai Wang, Li Chen

**Affiliations:** 1School of Agriculture, Yangtze University, Jingzhou 434025, China; 201871393@yangtzeu.edu.cn (M.Z.); 201972406@yangtzeu.edu.cn (Z.C.); 201972416@yangtzeu.edu.cn (N.Z.); xieguanglin@yangtzeu.edu.cn (G.X.); 2Institute of Life Science and Green Development, College of Life Science, Hebei University, Baoding 071002, China

**Keywords:** plant volatiles, dark black chafer, GC-EAD, host location, field trapping, hexanal, *β*-caryophyllene

## Abstract

**Simple Summary:**

The dark black chafer, *Holotrichia parallela*, which is widely distributed all over the world, is an economically important pest in agriculture and forestry. In the north part of China, this beetle causes serious damage to the peanut plant. Much attention has been paid to olfactory perception of volatile compounds from supplemental nutrition hosts by *H. parallela* prior to sexual maturation. However, volatile compounds attractive to this beetle from the peanut plant have not been identified yet. In this study, we collected the volatile compounds from peanut seedlings by dynamic headspace adsorption and identified twelve electrophysiologically active compounds responsible for the attraction of *H. parallela* to the peanut. Among the eight chemically identified compounds, *β*-caryophyllene and hexanal significantly attracted both sexes of *H. parallela* when tested individually in the field. A blend of *β*-caryophyllene and hexanal at a ratio of 2:1 was most attractive to the beetles. The addition of the remaining compounds to the binary mixture did not increase the attractiveness. The findings of this study reveal that *β*-caryophyllene and hexanal can be potentially used for development of effective attractants for management of *H. parallela*.

**Abstract:**

*Holotrichia parallela* (Coleoptera: Scarabaeidae: Melolonthinae) is a notorious pest of many crops, especially peanuts. In this study, volatiles from peanut plants were analyzed using both gas chromatographic-electroantennographic detection (GC-EAD) and gas chromatography/mass spectrometry (GC/MS) techniques, and tested for adult attraction with field trapping bioassays in Hebei Province, China. GC-EAD analyses indicated that *H. parallela* antennae strongly responded to twelve GC peaks, including eight identified compounds, (*Z*)-*β*-ocimene, hexanal, 6-methyl-5-hepten-2-one, nonanal, dihydromyrcenol, linalool, *β*-caryophyllene, methyl salicylate, and four unidentified compounds. When tested individually in field conditions from 24 to 31 July, 2020, *β*-caryophyllene and hexanal significantly attracted both sexes of *H. parallela*, whereas all other compounds were unattractive. A blend of *β*-caryophyllene and hexanal at a ratio of 2:1, close to the natural ratio of these two compounds from the intact peanut plant, was most attractive to the beetles. The remaining identified compounds, (*Z*)-*β*-ocimene, 6-methyl-5-hepten-2-one, nonanal, dihydromyrcenol, linalool, and methyl salicylate had no synergistic effects on *H. parallela* attraction when tested in combination with the blend of *β*-caryophyllene and hexanal. These results demonstrated that *β*-caryophyllene and hexanal in the volatiles from peanut plants have strong attraction to *H. parallela*. These two compounds have the potential to be used for monitoring *H. parallela* and its management programs.

## 1. Introduction

The dark black chafer, *Holotrichia parallela* Motschulsky (Coleoptera: Scarabaeidae: Melolonthinae), is one of the most economically important pest species in agriculture and forestry in China, Japan, Korea, and the Russian Far East region [1]. The scarab larvae live underground and feed on a large amount of plant roots, while the adults feed on the aerial parts of various plant species, such as corn, soybean, and other crops and trees. The beetle imposes a serious threat to peanut production in North China. The larvae (so-called “grubs”) feed on peanut roots and pods, resulting in 10–20% losses annually [2]. The adults hide in soil during the day, and come out at night to search for leaves, flowers, and fruits from crops for feeding. Spraying synthetic insecticides in the evening, while adults are actively feeding, could be an effective method. However, repeated applications of highly toxic insecticides to target grubs and adults can lead to environmentally detrimental consequences, i.e., the “3R” problems: residue, resistance, and resurgence [3]. Therefore, it is necessary to adopt an environmentally friendly solution for the management of *H. parallela*.

Volatile compounds released from host and non-host plants play a critical role in regulating the behavior of herbivorous insects, such as orientation, nutrition and mate location, oviposition, aggregation, and dispersal [4,5,6,7]. Adults of *H. parallela* can damage peanut leaves, and fertile females prefer to lay eggs in peanut fields. When searching for feeding, mating, and oviposition sites at nighttime, adult scarabs rely on olfactory cues from host plants. Olfactory cues can be volatile compounds from peanut seedlings. Research in chemical communication between plants and insects provides opportunities for the use of plant compounds against insect pests. Attractants originated from plant volatiles have been used for population monitoring and mass trapping [8,9]. The study on the Japanese beetle, *Popillia japonica* Newman (Coleoptera: Scarabaeidae: Rutelinae), is a successful example in developing an effective attractant based on volatiles of plant origins. A commercialized product of phenethyl propionate + eugenol + geraniol (3:7:3) is a superior attractant for *P. japonica* [10]. A floral attractant consisting of 3-methyl eugenol/1-phenylethanol/(*E*)-anethol/(±)-larvandulol has been used for field trapping of two rose chafers, *Cetonia aurata aurata* L. (Coleoptera: Scarabaeidae: Cetoniinae) and *Potosia cuprea* Fabr. (Coleoptera: Scarabaeidae: Cetoniinae) [11]. A mixture of (*Z*)-3-hexen-1-ol, geraniol, eugenol, and 2-phenylethyl propionate can attract the garden chafer, *Phyllopertha horticola* L. (Coleoptera: Scarabaeidae: Rutelinae), over a whole flight season [12]. Racemic 2,3-butanediol from sorghum is a powerful attractant for sorghum chafer, *Pachnoda interrupta* Olivier (Coleoptera: Scarabaeidae: Cetoniinae) [13].

In this study, we collected the volatile compounds from peanut seedlings by dynamic headspace adsorption and employed gas chromatograph-coupled electroantennographic detection (GC-EAD) and gas chromatography/mass spectrometry (GC/MS) analyses to identify active compounds responsible for attraction of *H. parallela* to peanut field. The electrophysiologically active compounds were then selected to test for field catches of *H. parallela* adults. The results would be useful in the development of effective attractants for future control efforts, either in mass trapping or in monitoring.

## 2. Materials and Methods

### 2.1. Insects and Plants

Adults of *H. parallela* were collected from an orchard (E116.83, N38.15) in 2019, in Zhangguantun, Cang County, Cangzhou, Hebei Province, China. The beetles were kept in clear plastic boxes (62 × 42 × 30 cm) with elm leaves and moisturized soil, and the culture was maintained at 26 °C, 60% relative humidity (RH), with a 12 h L: 12 h D photoperiod in a growth chamber. The elm leaves were replaced with fresh ones every 48 h. Peanut seeds (Cultivar Huayu 23) were planted in fertilized commercial soil (Shenxian Yuxinfeng Matrix Cc., Ltd., Baoding, China) in a plastic pot (diameter 22 cm, 13 cm deep) and the plants were grown in a plant growth chamber under the same conditions. Peanut plants in the pod bearing period were used in the plant volatile collection.

### 2.2. Headspace Plant Volatile Collection

Volatile collections were performed from 7:00 p.m. to 9:00 p.m. in June 2019, when the adults were active in the field. The aerial portion of the potted healthy peanut (*Arachis hypogaea* L.) plants was enclosed in a polyethylene (PE) bag (35 × 45 cm; Glad, Clorox China Limited, Guangzhou, China) and sealed tightly with cotton string [14]. Charcoal filtered continuous airflow at 550 mL/min was pumped into the bag using a vacuum pump (QC-1S; Beijing Municipal Institute of Labour Protection, Beijing, China) to maintain positive pressure. Half of a collection filter containing Porapak-Q (200 mg, 80/100 mesh, Supelco, Bellefonte, PA, USA) packed in a glass column (I.D. 5 mm) was inserted in the PE bag and tied tightly with a cotton string. Air was drawn out (500 mL/min) from the bag through the collection filter with a second vacuum pump. All parts of the system are connected by tetrafluoroethylene tubes. Prior to volatile collection, the filter trap was cleansed sequentially with methanol and dichloromethane (CNW Technologies GmbH, Duesseldorf, Germany) (3 mL each) and conditioned under a N_2_ flow (ca. 10 mL/min) at 150 °C for 30 min in an oven. Immediately after collection, the volatiles were eluted from the trap with 1 mL of dichloromethane [14]. Eight collections from different plants were carried out over a period of 2 h. The extracts were analyzed by gas chromatography (GC) and then combined and concentrated to 500 μL under a mild N_2_ stream, and then kept at 4 °C for later use. The odors in the empty PE bags were collected using the system described above as the control to exclude possible contaminants.

### 2.3. GC and GC-EAD

GC-EAD recordings were carried out with a Shimadzu GC-2010plus equipped with a flame ionization detector (FID) and a DB-WAX capillary column (30 m × 0.25 mm × 0.25 μm, Agilent Technologies). The experimental details were described by Chen et al. [14]. Briefly, injections (3 μL of sample, i.e., the dichloromethane solution of peanut plant volatiles eluted from the absorbent trap) onto the DB-WAX column were operated in a splitless mode with the split opened after 0.75 min. The oven temperature was programmed from 40 °C (isothermal for 2 min) to 120 °C at 5 °C/min, then to 240 °C at 15 °C/min, and held for 4 min, using nitrogen as a carrier gas at 2 mL/min. The injection port and detector temperatures were set at 230 °C and 250 °C, respectively. The GC effluent as well as a 20 mL/min N_2_ make-up gas was split at a ratio of about 1:2 with a glass push-fit “Y” splitter (Agilent Technologies). The EAD effluent was delivered into a stream of charcoal-filtered and humidified air (400 mL/min) over the antennal preparation.

The antenna of the beetle was excised and inserted between two glass capillaries (Outer Diameter 1.5 mm, Inter Diameter 0.84 mm, Vital Sense Scientific Instruments Co. Ltd., Chengdu, China) filled with Beadle–Ephrussi Ringer solution modified by Tween^®^ 80 (0.05%, *W*/*V*) [14]. The antenna base was inserted into the reference electrode, and one of the three lamellae was inserted into the recording electrode. The antennal signal was amplified 10 times, converted to a digital signal by IDAC-2, and recorded simultaneously with the FID signal on a computer using GC-EAD software (GC/EAD, version 1.2.4 or 4.4, Syntech, Kirchzarten, Germany). At least five successful GC-EAD runs were obtained for each sex of adult beetles, and traces were overlaid on the computer monitor to determine GC peaks that consistently yielded EAD responses.

The equipment and temperature program of GC analysis was the same as GC-EAD. The FID signal was recorded on a computer using GC-solution software (version 2.3, Shimadzu, Kyoto, Japan).

### 2.4. Chemical Identification

The headspace volatile samples were analyzed on an Agilent 7890A GC coupled to a 5975C Mass Selective Detector (GC/MS) with electron impact (EI) ionization mode at 70 eV. The GC was equipped with a DB-WAX capillary column as described above. The carrier gas was helium at an average linear flow rate of 1 mL/min. One microliter of each sample was injected in splitless mode with the split closed for 0.75 min at an injector temperature of 230 °C. The oven temperature was programmed as described above for GC-EAD. The temperature of the transfer line was set at 250 °C. The EAD active compounds were tentatively identified by comparison with the NIST08 MS library, and the retention times and mass spectra of identified compounds were confirmed through injecting synthetics under the same GC/MS program as described above. They were also verified by GC-EAD analyses on female and male antennae, injecting a mixture containing 100 ng of each authentic standard. All of the synthetic compounds were purchased from Sigma-Aldrich and had purities of 98–99%, except for *β*-ocimene (90%) and nonanal (95%).

### 2.5. Field Trapping 

The field experiments were conducted in the rural area of Cang County, Cangzhou City, Hebei Province (E116.82, N38.17) during mid-July to late August 2020, targeting the areas of corn and elm fields heavily infested by *H. parallela* adults in previous years. Cross-pane funnel traps (Pherobio Technology Co., Ltd., Beijing, China) (50 × 30 cm for each pane) were used in all three experiments, deployed in lines with 5 m between traps. The pure synthetic compounds and their mixtures were directly applied on dental cotton rolls, and placed into PE bags (5 × 7 cm; Jiutian Plastic Industry Co., Ltd., Huaining, China). The PE bags were then fixed to the lower end of the cross-pane of the traps. Ten replicates were separated by at least 50 m, depending on the topography. The position of the treatments was randomized within replicates and kept unchanged during the experiment. Traps were emptied daily and the beetles were counted and sorted by sex. Empty, unbaited traps were used as a control.

Experiment 1 was conducted from 24 to 31 July, 2020, testing the attraction of individual compounds identified from the peanut headspace volatiles. *β*-Ocimene, hexanal, 6-methyl-5-hepten-2-one, nonanal, dihydromyrcenol, racemic linalool, *β*-caryophyllene and methyl salicylate, were applied to dental cotton rolls at a dose of 50 mg each.

Experiment 2 tested the effect of different ratios of *β*-caryophyllene to hexanal (1:2, 1:1, 2:1, 4:1, 8:1) on beetle catch from 2 to 17 August, 2020. The total amount of each blend was 300 mg.

Experiment 3 was carried out from 13 to 27 August, 2020, to test the attraction of blends of above compounds in different combinations. The blend was comprised of *β*-caryophyllene and hexanal, and one of the remaining compounds, at a total amount of 300 mg. The proportions of each compound in a blend were in a ratio mimicking what was found in the peanut headspace.

### 2.6. Statistical Analyses

Trap catch data among the treatments conformed to Poisson distribution and were analyzed by Poisson regression followed by Bonferroni-adjusted pairwise comparisons (SPSS 20.0) at α = 0.05.

## 3. Results

### 3.1. Identification of Active Host Plant Volatiles

At least 12 reproducible EAD responses in female and male antennae of *H. parallela* were observed to volatiles from intact peanut plants (Figure 1A). The EAD active compounds were identified as hexanal, (*Z*)-*β*-ocimene, 6-methyl-5-hepten-2-one, nonanal, dihydromyrcenol, linalool, *β*-caryophyllene, and methyl salicylate, and the remaining four compounds were unknown. GC-EAD runs using synthetic standard solution (0.1 μg/μL for each compound) confirmed electrophysiological activity of the identified compounds (Figure 1B).

### 3.2. Field Evaluation of Synthetic Compounds

In experiment 1, a total of 94 beetles (54 females and 40 males) were caught from 10 replicates of 9 chemical traps. *β*-Caryophyllene and hexanal attracted significantly more beetles than all other chemicals (Figure 2A; Poisson regression, *p* < 0.001, *χ*^2^ = 29.872, *df* = 8 (Bonferroni corrected)). Moreover, there were no significant differences between other chemicals and the blank. However, the number of beetles trapped by hexanal was marginally greater than that by dihydromyrcenol and methyl salicylate (significant at α = 0.1, *p* = 0.081 (Bonferroni corrected)).

A total of 107 beetles (65 females and 42 males) were trapped in the second experiment. The trap baited with the *β*-caryophyllene/hexanal blend at a ratio of 2:1 captured significantly more beetles than all of the other ratios and blank (Figure 2B; Poisson regression, *p* < 0.001, *χ*^2^ = 42.195, *df* = 5 (Bonferroni corrected)). Significantly more beetles responded to the ratio 1:2 than to the control. Other ratios, 1:1, 4:1, 8:1, captured relatively more beetles than blank, but there was no significant difference between these treatments and control (Figure 2B).

In experiment 3, a total of 114 beetles (64 females and 50 males) were captured from 10 replicates of 8 treatments. Significantly more beetles were attracted to the *β*-caryophyllene/hexanal blend than to the blank control and all of the ternary blends, but the mixture of *β*-caryophyllene, hexanal, and nonanal (Figure 2C; Poisson regression, *p* = 0.01, *χ*^2^ = 25.831, *df* = 7 (Bonferroni corrected)). Relatively fewer beetles responded to the ternary blend of *β*-caryophyllene, hexanal, and nonanal than to the binary blend, but the difference was not statistically significant (Figure 2C). 

## 4. Discussion

Plant volatiles can act as important chemical signals that form communication links between insect herbivores and their host plants and function as either attractants or repellents [15,16]. In this study, we investigated which peanut volatile components play a role in host plant selection in *H. parallela*. Our GC-EAD and GC/MS analyses found twelve peanut volatile compounds, (*Z*)-*β*-ocimene, hexanal, 6-methyl-5-hepten-2-one, nonanal, dihydromyrcenol, linalool, *β*-caryophyllene, methyl salicylate, and four unidentified compounds (Figure 1), which are consistently detected by the antenna of *H. parallela* adults.

In a previous study, 19 components were identified in healthy peanut plant volatiles, including *E*-2-hexenal, *Z*-3-hexen-1-ol, *α*-pinene, *β*-pinene, myrcene, limonene, (*Z*)-*β*-ocimene, linalool, DMNT, (*Z*)-3-hexenyl isobutyrate, (*Z*)-3-hexenyl butyrate, indole, (*Z*)-jasmone, *β*-caryophyllene, *α*-bergamotene, *α*-humulene, *β*-farnesene, nerolidol, and TMTT [17]. Among these compounds, (*Z*)-*β*-ocimene, linalool, and *β*-caryophyllene were found to elicit antennal response in *H. parallela* adults in the present study. Volatile emission in peanut plants showed a diurnal pattern, but Cardoza et al. [17] only presented data with peak release during the afternoon. Our volatile collections were conducted during the nighttime when the scarab beetles were actively feeding. It seems likely that healthy peanut plants release the five compounds, not reported in the previous study, hexanal, 6-methyl-5-hepten-2-one, nonanal, dihydromyrcenol, and methyl salicylate, only under a dark phase. The maximum release of these electrophysiologically active volatiles by peanut plants may coincide with the period of the beetle’s greatest flight activity. We assume that these volatiles, emitted in relatively large amounts during the period of beetle flight, may serve as a reliable guide to conspecifics and suitable host plants. It is therefore critical to perform host plant volatile collections when the beetles are actively searching for host plants.

The electrophysiological and behavioral activities of these compounds have been reported in earlier studies. Hexanal from maize was electrophysiologically active in antennae of *H. parallela* female adults [18]. Among the 10 compounds isolated from *H. parallela* host plants [(*Z*)-3-hexen-1-ol, (*E*)-2-hexen-1-ol, (*Z*)-3-hexenyl acetate, (*E*)-2-hexenyl acetate, (R)-(+)-limonene, α-phellandrene, α-pinene, (*Z*)-*β*-ocimene, methyl benzoate, and benzaldehyde], (*E*)-2-hexenyl acetate, and (*Z*)-3-hexenyl acetate were found to elicit the strongest EAG responses in both female and male *H. parallela* [2]. (*Z*)-3-Hexenyl acetate elicited much higher EAG response in both female and male beetles compared to *β*-caryophyllene and *a*-phellandrene, and the EAG response to *β*-caryophyllene was significantly higher in female antennae than in male antennae [19]. These host plant volatiles have been found to bind to odorant-binding proteins (OBPs) of *H. parallela*. (*Z*)-3-Hexenyl acetate showed a high binding affinity to HparOBP-1, HparOBP-2, HparOBP20, and HparOBP49. HparOBP-1, HparOBP-2, and HparOBP49 specifically bound to general odorants and green leaf volatiles, whereas HparOBP20 showed a broad spectrum of binding activity to plant volatiles [19,20]. Among the identified EAD-active volatiles from the peanut plant, racemic linalool bound only to HparOBP-2, and methyl salicylate to HparOBP20. The remaining EAD-active volatiles did not bind to these OBPs, or their binding affinities were not tested. In Y-tube bioassays, females displayed the highest selective response rates to (*E*)-2-hexenyl acetate and (*Z*)-3-hexenyl acetate, and males had a significantly higher selective response rate only for (*E*)-2-hexenyl acetate [2,19]. *Potosia brevitasis* Lewis (Coleoptera: Scarabaeidae: Cetoniidae) adults had a strong EAG response to nonanal [21]. In a Y-tube olfactometer test, racemic linalool and *β*-caryophyllene had significant attractive effects on *Popillia quadriguttata* Fabricius (Coleoptera: Scarabaeidae: Rutelinae) females [22]. Racemic linalool can induce EAG and behavioral response of *Holotrichia oblita* Faldermann (Coleoptera: Scarabaeidae: Melolonthinae) [23]. Previous field experiments indicated that methyl salicylate commonly found in the odor profile of flowers and fruits have potential as attractants for sorghum chafer, *P. interrupta* [24].

In this study, individual *β*-caryophyllene, hexanal, and their mixture attracted more beetles than other treatments in field trapping experiments (Figure 2A,B). Generally, mixtures are more attractive than single compounds, which may be used for pest control [25,26]. For instance, a trap baited with methyl salicylate-eugenol blend caught significantly more sorghum chafers than traps baited with the single compounds [13]. Furthermore, the ratio of compounds in the blend is also an important factor affecting its attractiveness. In order to determine the optimum ratio of the blend of *β*-caryophyllene and hexanal, the attractancy of varying ratios of *β*-caryophyllene to hexanal were tested on beetle catch. The results indicated that the *β*-caryophyllene/hexanal blend at a ratio of 2:1 was the most attractive to *H. parallela* (Figure 2B). Subsequently, we used the blend at 2:1 to prepare ternary mixtures for further field experiment.

The addition of a third compound to the binary mixture did not increase its attractiveness to *H. parallela* (Figure 2C). It is possible that the third compound may be active when mixed with other volatile compounds. Furthermore, it is not clear what specific roles the four unidentified compounds in this study play in peanut location by *H. parallela*. Identification of these compounds and evaluation of their attractiveness in combination with the *β*-caryophyllene/hexanal blend awaits further study. A recent study indicated that adults of *H. parallela* showed significant preference to benzyl alcohol derived from its non-host *Ricinus communis* L. (Euphorbiaceae) [27]. It is worthy of testing the attractiveness of a combination of benzyl alcohol and the *β*-caryophyllene/hexanal blend.

The relatively low trap catches in this study may attribute to trap surroundings. As we performed field trials in cornfields, volatiles emitted from surrounding crops might include some of the compounds tested, or similar compounds, which could affect the results of field experiments [28]. These surrounding natural host plants appear to outcompete lures in the traps. The annual dark black chafer peaks occur during mid-June to mid-July. The beetle population started declining after mid-July, and was probably present at a relatively low level during our trapping period.

(*Z*)-3-Hexenyl acetate is a common herbivory-induced plant volatile and often plays an important role in insect-plant interactions [29]. More males of *H. parallela* were attracted than females to (*Z*)-3-hexenyl acetate in a field test [19]. This compound, released in large amount by peanut plants when exposed to feeding by *Spodoptera exigua* Hübner (Lepidoptera: Noctuidae), can be incorporated in the *β*-caryophyllene/hexanal blend in future field tests. In addition, plant volatiles have proven to act as synergists for sex pheromones [2,6,30]. Two host-plant volatile compounds, (*Z*)-3-hexenyl acetate and 1-undecanol, increase sex pheromone attraction of *Grapholita molesta* (Busck) (Lepidoptera: Tortricidae) [31]. The mixture of (*E*)-2-hexenyl acetate and the sex pheromones of *H. parallela* is significantly more attractive than the sex pheromones alone [2]. It seems likely that the *β*-caryophyllene/hexanal blend can synergize the attractiveness of the sex pheromones of *H. parallela*.

## 5. Conclusions

The present study showed that *β*-caryophyllene and hexanal were the most attractive compounds among the electrophysiologically active compounds from peanuts, and the binary blend of *β*-caryophyllene and hexanal was more attractive than all ternary blends. The optimum ratio of *β*-caryophyllene to hexanal was 2:1, which is similar to their natural ratio. As peanut plants show an arrestment effect on them [32], female beetles of *H. parallela* may land for oviposition in the vicinity of peanut roots after perceiving these two volatile compounds emitted from this suitable ovipositional host plant. These findings may offer valuable information for developing new attractants for *H. parallela* control. Furthermore, the *H. parallela* beetle is a strong flyer at nighttime and highly mobile, and its flight periods peak during mid-June and late July. The adults search for host plant leaves for feeding at sunset and return to soil at dawn [33,34], suggesting an important role of volatile compounds released by host plants in this process. It may serve as a useful model for future research on host searching in foliage-feeding insects. It is interesting to note that *β*-caryophyllene is attractive to many insects e.g., [35,36,37], but toxic to some other insects e.g., [38,39]. Further studies are needed to interpret why there are different modes of action of this compound to different insects.

## Figures and Tables

**Figure 1 insects-12-00158-f001:**
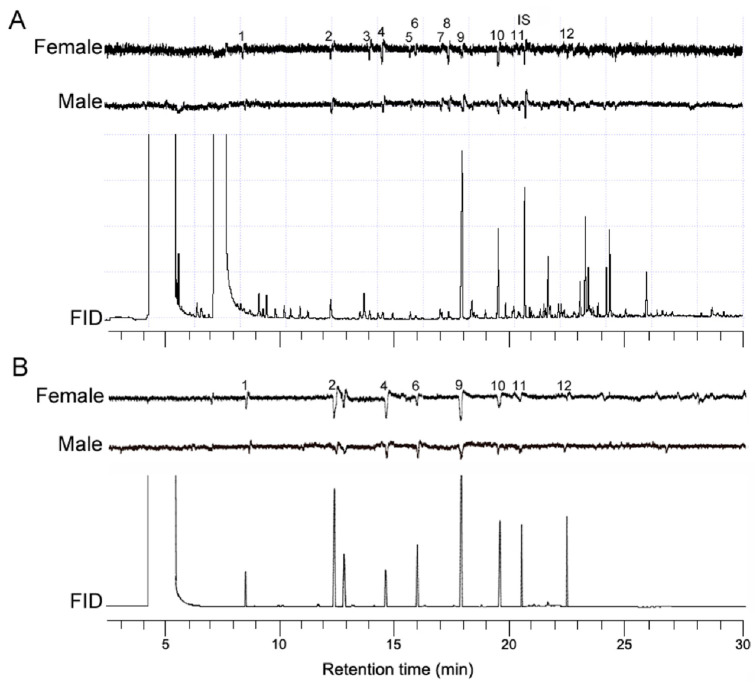
Simultaneous responses of flame ionization detector (FID) and of *H. parallela* antenna (electroantennographic detection, EAD). The upper traces in each figure represent the signals from the EAD and the lower trace represents the signal from the FID. (**A**) Headspace volatile sample collected from the peanut plants. (**B**) Mixture of synthetic compounds. The upper trace in each figure represents the signal from the EAD and the lower traces represent the signals from the FID. 1, hexanal (8.35; 3.04%) *; 2, (*Z*)-*β*-ocimene (12.31; 2.54%); 3, unidentified (13.69; 7.50%); 4, 6-methyl-5-hepten-2-one (14.38; 5.80%); 5, unidentified (15.45; 7.89%); 6, nonanal (15.76; 5.47%); 7, unidentified (16.75; 9.74%); 8, unidentified (17.12; 6.06%); 9, dihydromyrcenol (17.76; 29.20%); 10, linalool (19.46; 12.15%); 11, *β*-caryophyllene (20.40; 5.46%); 12, methyl salicylate (22.48; 5.14%); IS, methyl benzoate. * Retention time and relative percentage of the EAD active compounds.

**Figure 2 insects-12-00158-f002:**
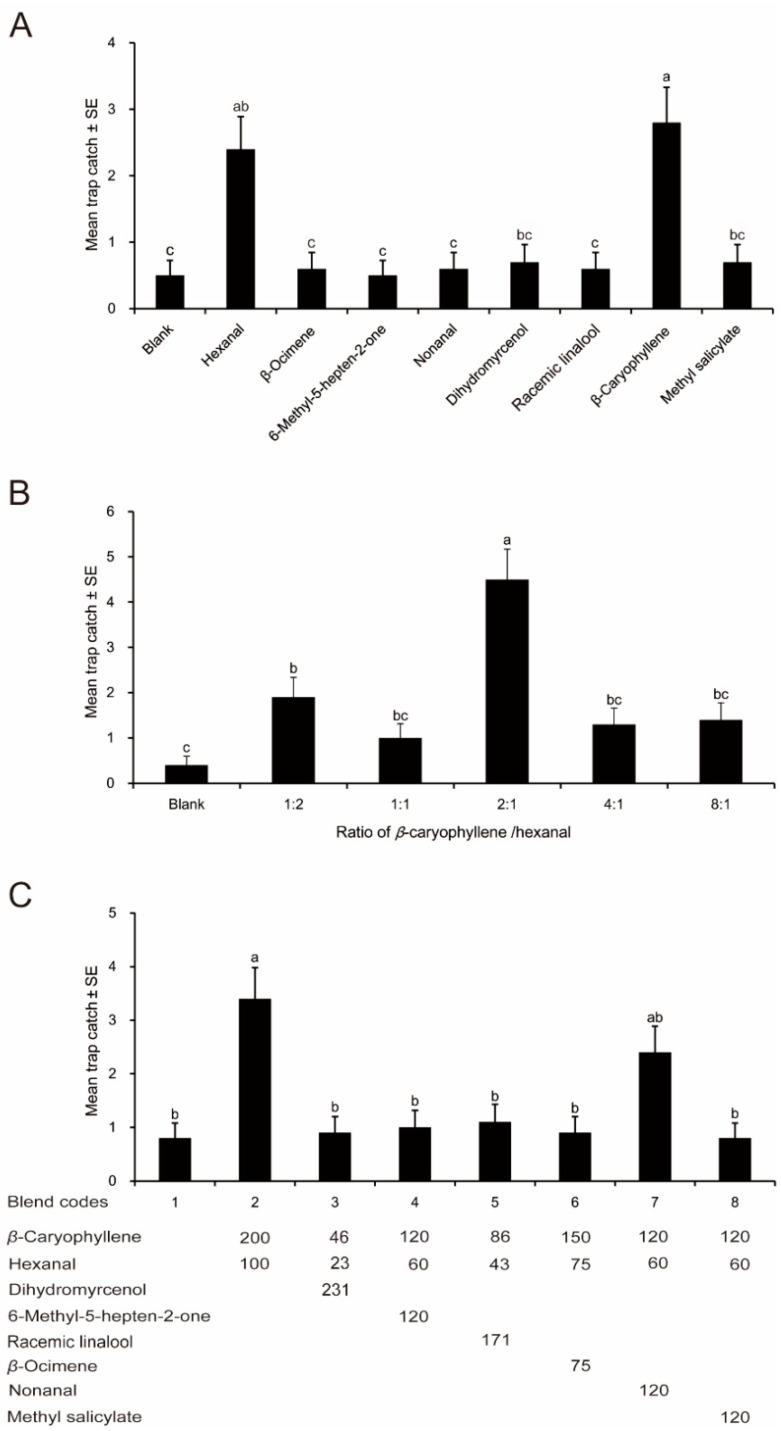
Adults of *H. parallela* captured in traps with synthetic compounds in Cangzhou, Hebei Province, China. Columns with different lowercase letters are significantly different at α = 0.05 (N = 10, Poisson regression with Bonferroni correction). (**A**) Individual synthetic compounds; (**B**) Differing *β*-caryophyllene/hexanal ratios; (**C**) *β*-Caryophyllene/hexanal blend plus differing plant volatile components. Blend codes are the code numbers of treatments, the values under the blend code indicate the amount of each compound in a treatment.

## Data Availability

The data presented in this study are available on request from the corresponding authors.

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
