# Peer review of "Electrophysiological and Behavioral Responses of Holotrichia parallela to Volatiles from Peanut"

_insects, 2021, doi:10.3390/insects12020158_

Round 1

Reviewer 1 Report

This is a review of the manuscript entitles “Electrophysiological and behavioral responses of Holotrichia parallela to volatiles from peanut”. The manuscript has an interesting subject, sound methods, acceptable analyses, novelty, and clarity. However, it should be revised based on the following issues;   

Line 12: Is peanut one of the supplemental nutrition hosts? In the previous sentence, it is introduced as the main host.

Lines 35-37: rewrite the conclusion. Are you investigate the host-plant location of H. parallela? Have you checked all volatiles? Refer to the “attraction” of β-Caryophyllene and hexanal.

Line 45: I think reference 2 (Shan et al.) has not relevant to the economic importance of the insect pest. 

Lines 51-51: This sentence is incorrect. For example, spraying trap-plants or hosts with synthetic insecticides in the evening can be an effective method.

Line 54: did you used the relevant references for 3R problems? None of these references are related to residue, resistance, and resurgence before chemical insecticides. As a reader, I want to read about this sentence. Is this possible with these references? Check all references.

Line 58: used “nutrition” instead of “food”.

Line 61: more explain the application attractants. It is one of the main strategies in your work.

Line 73: add a space before “and”. Be consistent. In the previous scientific names of insects, the common names were added but not here.

Line 79: remove the sentence. It was explained at the end of the paragraph.

Line 91: write the geographical location of the orchard.

Line 92: write the ratios. Is there a right reference for these materials and their ratios? Did you have any fluctuations in temperature and humidity?

Line 94-96: It is necessary to write details of the pots, including their size and material.

Line 99: … adults were active in the field. It's excellent, but why peanut was planted in the pods?

Line 100: which cultivar?

Line 106: How did you measure the exhaust air?

Line 111: add a reference for elution of volatiles by dichloromethane. Could you use other compounds?

Line 119: is the reference was written according to journal instruction?

Line 120: have you used the pure samples or added any solvent?

Line 149: How did you identify male and female insects?

Line 154: write the geographical location of rural areas.

Line 174: in what proportions were the compounds blended?

Fig. 2A and line 205: based on the error bars of columns, there is not any significant difference between beta-ocimene and linalool, but why they have different letters? Is there any significant difference between dihydromyrcenol with others?

Line 227: the repellent effects of plant volatiles were not assessed in the study of Donaldson et al. (18). Add a relevant reference. I reemphasize that you should check the references for correct correlation with the sentences used.

Line 229: location or selection?

Line 232: the insecticidal effects of β-ocimene, linalool, and β-caryophyllene had been reported by several studies. Didn't you detect any mortality in the captured adults?

Why is β-caryophyllene toxic and repellent to some insects and attracting to some others such as H. parallela? No discussion has been written in this regard. This should be explained comprehensively.

Line 327: model or method?

References: Be consistent and write the abbreviated or full names of journals according to journal instructions.

Author Response

Dear Reviewer 1,

We sincerely thank you for valuable comments that significantly improve our manuscript. Following is our responses to the comments.

Best Regards,

Zhang Mengmeng, on behalf of all coauthors 

Reviewer 2 Report

This manuscript using GC-MS and GC-EAD to analysis the active volatiles from peanut seedlings. The author get 12 active compounds including four unknown odorant. Through field trap, the experiments were designed to test the odorants attraction to H. parallela. From single compound test to recombination test, the experiment is designed logical and optimal for odorant selection. Some questions as follow also need the author to give response.

  1. In the material section, in the experiment 3, the blend was comprised of β-caryophyllene and hexanal plus one of the remaining compounds. I also confused about the ratio of each compound, and the author need give the detail in this part.

  1. In the field trap of the material section, the author showed “The synthetic compounds and their mixtures were directly applied on dental cotton rolls”, I am confused about the compound use and need the author to determine the compound is directly used not dissolve in the solvent, and then use. If used the solvent, need to complement the concentration.

  1. The three experiments were conducted in different time, why the experiments were not designed in the same time?

  1. In the result section 3.1, the author found 12 compounds had EAG response, among them, eight compounds were identified, I suggest the author can further analysis the result and showed which compound respond well in different sex.

  1. In figure 2, Why the author use different amount of third compound blend withβ-caryophyllene and hexanal? Why not use the same amount to blend with β-caryophyllene and hexanal?

Author Response

Dear Reviewer 2,

We sincerely thank you for valuable comments. We carefully went through these comments. Following is our responses to the comments.

Best Regards,

Zhang Mengmeng, on behalf of all coauthors

Reviewer 3 Report

The manuscript by Zhang et al. describes a generally thoroughly undertaken work. I have the following comments:

  • Please provide a better rationale for this work. As far as I am aware, there is a sex pheromone synergist for H. parallela. I understand the peanut volatiles are good candidates for a new synergist, but this needs to be better explained in the introduction.
  • line 45: I think H. parallela is only a pest of peanut in East Asia!
  • line 45: Provide better references for distribution (e.g. CABI).
  • lines 51-52: Complete sentence - Why are insecticides not efficient?
  • line 103: Provide exact volume of air inflow.
  • line 148: Be more precise about peak confirmation using synthetic standards: What method was used?
  • line 168: Do you mean rac. linalool?
  • lines 170-172: I understand the use of 1:2 β-caryophyllene-to-hexanal ratio, but how did you decide on the other ratios?
  • line 185: (E)- or (Z)-beta-ocimene?
  • line 186: correct to `dihydromyrcenol`
  • Fig. 1B: Put corresponding numbers on GC-EAD trace to indicate peak identities!
  • lines 313-315: Please expand on why you think the binary blend might synergise the attractiveness of the sex pheromone.

Author Response

Dear Reviewer 3,

We sincerely thank you for valuable comments. W e carefully went through these comments and revised the manuscript accordingly. Following is our responses to the comments.

Best Regards,

Zhang Mengmeng, on behalf of all coauthors

Reviewer 4 Report

This is an interesting manuscript on the antennal and behavioral responses of H. parallela to peanut plant volatiles.

The reviewer has the following comments:

Line 210. “to the ratio 1:2 than to the control.”

Figure 2 Please provide more explanation of the blend code numbers, 200, 100, 60 30 etc.

Line 240-245. Is there any known physiological or ecological explanation for the timing of different releases of the five compounds only to the dark phase? Is it known what parts of the plant release these compounds?

Line 406 References 1 and 32 are the same.

Author Response

Dear Reviewer 4,

We sincerely thank you for valuable comments that significantly improve our manuscript. We carefully went through these comments and revised the manuscript accordingly. Following is our responses to the comments.

Best Regards,

Zhang Mengmeng, on behalf of all coauthors

Round 2

Reviewer 1 Report

Dear,

Despite the corrections and explanations made, the following issues should be addressed before acceptance of the manuscript.

Line 37: add a dot before "these two compounds". Check the manuscript for such mistakes.

Lines 44-46: the words you added have disrupted the sentence structure. Write this sentence more carefully. 

Line 78-79: add the common name of pests in the first mention, like the other names you wrote. 

Line 98: what proportion of the soil and the leaves were poured into plastic boxes?

Line 101: write very briefly in parentheses what you mean by "nutritional soil". Did you add anything to the soil? 

Line 127: mention the answer you gave in the text; The sample was the DCM solution of peanut plant volatiles eluted from the absorbent trap.

Line 156: what is "ofng"?

Fig. 2A: check the results of the comparison of means again. Where is the b-column? Also, I think dihydromycenol and methyl salicylate have not significant difference with linalool.

Add a recommendation for further works at end of the discussion about your explanation; β-Caryophyllene is attractive to many insects. Although it is also toxic to some insects, the goal of our study was to develop attractants from peanut volatiles. New study is needed to interpret why there is different mode of action of this compound to different insects. We will work on this in the near future.

Author Response

Dear Reviewer 1,

We sincerely thank you for valuable comments that significantly improve our manuscript. We carefully went through these comments and revised the manuscript accordingly. Following is our responses to the comments.

Best Regards,

Li Chen

Reviewer 3 Report

Please see my comments to some of the author`s responses.

The airflow rate of the vacuum pump was set at 500 ml/min, and collection time was 2 h. We also measured flowrate with a flowmeter to confirm the accuracy of the pump prior to volatile collection.

This is still not clear to me. Do you mean the airflow going into the bags was set to 500 mL/min, and the airflow sucked out of the bags was also 500 mL/min? I wonder if this is right, because you mention `positive pressure` in the bags. Positive pressure means the bags are slightly over-pressurised compared to air pressure in the surrounding room, ensuring that unfiltered air cannot enter the sampled headspace. Please clarify.

We confirmed the retention time and EAD activity of the identified compounds with synthetics.

To confirm GC peak match between sample and synthetics, do you simply check retention times, or did you calculate retention indices, which are more accurate?

Yes, it is racemic linalool, not chiral.

It is fine to just say `linalool` when you talk about an identification without enantiomeric resolution, but where ever you mention linalool used as a SYNTHETIC compound in the text, please add `rac.` before the name.

Author Response

Dear Reviewer 3,

We sincerely thank you for valuable comments that significantly improve our manuscript. We carefully went through these comments and revised the manuscript accordingly. Following is our responses to the comments.

Best Regards,

Li Chen, on behalf of all coauthors
